# Comparison of Sweet–Sour Taste Interactions between Cold Brewed Coffee and Water

**Jonas Yde Junge [1,2,\*]** , **Line Ahm Mielby [3]** and **Ulla Kidmose [1,2]**

1    Food Quality Perception and Society Team, iSense Lab, Department of Food Science, Aarhus University, 8200 Aarhus, Denmark; ulla.kidmose@food.au.dk
2    Sino-Danish Center for Education and Research (SDC), 8000 Aarhus, Denmark
3    Danish Technological Institute, 8000 Aarhus, Denmark; lmie@teknologisk.dk
\*    Correspondence: jonas.junge@food.au.dk; Tel.: +45-26330862

**Abstract:** Most beverages are complex matrices. Different taste compounds within these matrices interact, and thus affect the perception of the tastes. Sweetness and sourness have generally been known to suppress each other, but often such investigations have focused on aqueous solutions. Investigations into what happens when these known interactions are transferred to more complex solutions are scarce. In this study, we investigated the differences in taste interactions between an aqueous matrix and a cold-brewed coffee matrix. Two sub-studies were conducted. In one, six aqueous samples were evaluated by 152 naïve consumers; in the other six cold-brewed coffee samples were evaluated by 115 naïve consumers. In both studies participants tasted samples with no addition or with addition of either sucrose, citric acid, tartaric acid, or a mix of sucrose and either of the acids. Results showed that the sweetness of sucrose was suppressed by both citric acid and tartaric acid in both matrices. The sourness of both citric acid and tartaric acid was suppressed in the aqueous matrix, but only the sourness of tartaric was suppressed in the coffee matrix. Generally, the suppression was lower in the coffee matrix compared to the aqueous matrix. In conclusion, results from taste interaction studies conducted on aqueous matrices can to some extent, with caution, be interpolated to more complex matrices. Importantly, suppression effects might diminish with an increase in matrix complexity.

**Keywords:** binary taste interactions; taste mixtures; mixture suppression; consumer study; sensory perception; food matrix

## 1. Introduction

Most beverages are complex matrices containing many different taste and flavor compounds [1,2]. Tastes interact and thereby influence the perception; thus, differences in one taste might change the perceived intensity of another taste. Many studies have investigated taste interactions in aqueous solutions or model matrices [3–5]. Early studies in taste interactions indicated that the interactions found in aqueous solutions were also present in more complex matrices [6–8]. Taste interactions generally shows the same tendencies between matrices, e.g., that suppression in an aqueous solution would also result in suppression in a more complex matrix. That said, the magnitude of differences in taste interactions between aqueous solutions and complex matrices has only seldom been investigated [2].

Many studies have investigated the interaction effect between sweetness and sourness, and most find that sweetness suppresses sourness and sourness suppresses sweetness [5,6,8–12], but exceptions are seen. Veldhuizen et al. [3] investigated taste interactions in a lemon-based model matrix. They found sourness to be suppressed by sucrose, but in this matrix citric acid did not affect sweetness.

The general picture of suppression has been found in different matrices such as water [5,6,9–12], model beverage matrices [11,13–15], and complex beverage matrices [7,11,16–18].

Zamora et al. [11] investigated sweet and sour taste interactions in an aqueous solution, a wine model, and a white wine. They found a higher suppression of sweetness by tartaric acid in the wine matrix than in the aqueous solution, whereas the suppression of sourness by sucrose in the wine matrix was lower than in the aqueous solution. Generally, results for the ethanol-containing model matrix showed interactions that were in between the two other matrices in magnitude.

As taste interactions depend on both concentration of taste stimulus and the quality of the taste stimulus [5,8], a different effect of different acids on taste interactions is possible. Most often, taste interaction studies use sucrose as sweetener and citric acid as the preferred acid [3,5,6,9], though tartaric [10,11], malic [9], and lactic acid [9] have been investigated. Besides its presence in wine, tartaric acid occur naturally in many fruits as well as coffee [19,20]. Further, it is used as a flavoring and additive in many foods and beverages [21].

Beverages such as coffee, beer, wine, carbonated soft drinks, and fruit nectars are complex matrices with many different compounds adding to the taste perception [1,2]. One of the main differences between aqueous solutions often studied and beverage on the market is the product complexity, enabling flavor interactions on both peripheral physiological and central cognitive levels [22,23]. Coffee is a complex matrix containing many different taste- and flavor compounds contributing to the sensory perception [24,25].

The present study aimed to investigate the effect of the liquid matrix on taste interactions between sweetness and sourness. Further, the aim was to investigate the difference in interaction between two different sour compounds, namely tartaric- and citric acid. By using sucrose as sweet stimulus and citric and tartaric acid as sour stimuli, in an aqueous and a cold-brewed coffee solution, respectively, the matrix effects of suppression of sweetness by sourness and the suppression of sourness by sweetness were investigated.

## 2. Materials and Methods

The taste interaction effects between sucrose and citric or tartaric acid were investigated using a $2 \times 3$ full factorial design resulting in six samples in total in two different matrices, water or cold-brewed coffee. The concentrations of citric acid and tartaric acid in the samples used in this study were found by Junge et al. [10] to be similar in sourness intensity when evaluated in water.

### 2.1. Samples

Samples consisted either of aqueous solutions or cold-brewed coffee. Added to these two matrices were either sucrose (Merck KGaA, Darmstadt, Germany), citric acid (Merck KGaA, Darmstadt, Germany), or tartaric acid (L-(+)-Tartaric acid, Sigma-Aldrich, Co., St. Louis, MO, USA), or combinations (Table 1). Water for the aqueous solutions was San Benedetto, still natural mineral water (San Benedetto S.p.A., Scorzè, Italy), and cold-brewed coffee was Löfbergs Antioquia Reserve Cold Brew (Peter Larsens Kaffe, Viborg, Denmark). Samples are named Control ie. either water or cold-brewed coffee (CON), Citric Acid (CA), Tartaric Acid (TA), Sucrose (SU), Sucrose, and Citric Acid (SUCA), and Sucrose and Tartaric Acid (SUTA).

20-mL samples were served in opaque sample tubes with red lids (Fisher Scientific, Roskilde, Denmark) coded with three-digit numbers. The samples were stored at 5 °C for one day prior to testing and evaluated at room temperature.

**Table 1.** Sample names and concentrations of taste stimuli in either aqueous solution or cold-brewed coffee. Samples are control (CON), Citric Acid (CA), Tartaric Acid (TA), Sucrose (SU), Sucrose, and Citric Acid (SUCA), and Sucrose and Tartaric Acid (SUTA).

| Sample | Sucrose (% *w/v*) | Citric Acid (% *w/v*) | Tartaric Acid (% *w/v*) |
|---|---|---|---|
| CON | - | - | - |
| CA | - | 0.140 | - |
| TA | - | - | 0.114 |
| SU | 2.50 | - | - |
| SUCA | 2.50 | 0.140 | - |
| SUTA | 2.50 | - | 0.114 |

*2.2. Consumer Studies*

Both sub-studies were conducted at Aarhus University, Aarhus, Denmark. The six aqueous samples were evaluated on 21 May 2019, while the cold-brewed coffee solutions were evaluated on 27 September 2019. Students and staff were recruited for the substudies, which were conducted in student eating areas, and only healthy adults were included. The characteristics of the participants from the two sub-studies can be found in Table 2. Participants who verbally agreed to participate in the study were asked to complete the questionnaire on their smartphone or a provided iPad. Participants received the six samples in a zip-lock bag and a plastic cup with water for rinsing the mouth after tasting the samples.

**Table 2.** Baseline characteristics of participants in aqueous solution and cold-brewed coffee sub-studies. Provided *p*-values of ANOVA for the difference between sub-studies. For age, range is in parenthesis, BMI is Body Mass Index.

| | Aqueous (n = 152) | Cold Brew (n = 115) | *p*-Value |
|---|---|---|---|
| Mean of Age | 23.7 (19–48) | 24.6 (18–63) | 0.17 |
| Number of Female | 97 (75.8%) | 65 (66.5%) | 0.28 |
| BMI | 23.6 | 23.1 | 0.27 |
| Sweet foods liking | 5.24 | 5.43 | 0.30 |
| Sour foods liking | 5.03 | 4.39 | <0.01 |

Participants were asked to evaluate one sample at a time and to make sure that the sample codes in the questionnaire matched the ones on the samples. Samples were evaluated in randomized order following a Williams Latin Square design, and all data were collected using the software Compusense (Compusense Inc., Guelph, ON, Canada). In the questionnaire, the participants were additionally instructed to rinse their mouth with water and wait at least 30 seconds between the samples. Participants in the aqueous substudy were rewarded with a piece of chocolate for participation, whereas participants in the cold-brewed coffee sub-study were rewarded with a cup of coffee.

The questionnaire contained both sample-related questions and non-sample-related questions. The sample-related questions were ratings of sourness and sweetness intensity evaluated on a 9-point scale with anchors "Not at all" and "Extremely". Liking was evaluated on a 9-point hedonic scale with anchors "Dislike extremely" corresponding to 1, "Neither like nor dislike" corresponding to 5 and "Like extremely" corresponding to 9. Lastly, Just About Right evaluations were conducted on a 5-point scale with "Too little" corresponding to 1, "Just about right" corresponding to 3, and "Too much" corresponding to 5. In the coffee sub-study, bitterness intensity and bitterness Just About Right were also evaluated. Non-sample-related questions were nationality, gender, age, weight, height, and a battery of four questions on a 7-point Likert scale concerning general sweet and sour foods liking ("I like to eat sweet foods", "I like to eat sour foods", "I like to drink fruit juices", "I like to drink soft drinks") ranging from "Strongly disagree" corresponding to 1

to "Strongly agree" corresponding to 7. The sub-study with cold-brewed coffee samples ended with a questionnaire about coffee knowledge.

Body Mass Index (BMI = weight (kg)/height (m)$^2$) was calculated based on participant height and weight.

### 2.3. Statistical Analysis

All data analyses were performed using R [26] and RStudio [27]. Plotting and data formatting was executed using the Tidyverse package [28] and reshape2 [29]. Baseline characteristics of the participants in each sub-study were analyzed using the R package tableone [30].

Means were calculated using the R package SensoMineR [31]. Product effects were analyzed using mixed-model ANOVA on product effects using R package SensMixed [32], with Mixed Assessor Model (MAM) to adjust for scaling, where participant effects were considered random effects. To identify the presence of sucrose or one of the acids in the analysis, dummy variables "acid" and "sucrose" was constructed. The effects of acid (tartaric or citric), sucrose, matrix, and the interaction between these on either sweetness or sourness were analyzed using mixed-model ANOVA with MAM.

Tukey's Honest Significant Differences test (HSD) was used to determine sample differences using the R package lmerTest [33]. Matrix differences were visualized in the lollipop chart using R packages rstatix [34] and emmeans [35]. P-values showed in the lollipop chart were adjusted using Bonferroni corrections.

Estimations of suppression of both sweetness- and sourness in the two matrices was conducted using the method proposed by Calvino et al. [2]. Thus, percentages of suppression were calculated as follows

$$SwP_m = (\frac{1}{n} \sum_{i=1}^{n} \frac{Sw_{pmi} - Sw_{cmi}}{Sw_{pmi}}) \cdot 100\% \tag{1}$$

and

$$SoP_m = (\frac{1}{n} \sum_{i=1}^{n} \frac{So_{pmi} - So_{cmi}}{So_{pmi}}) \cdot 100\% \tag{2}$$

where *SwP* is Suppression Percentage for Sweetness, *SoP* is Suppression Percentage for Sourness, *So* is sourness, *n* is number of subjects, *p* is pure (only sucrose), *c* is a combination of two taste stimuli (sucrose and an acid), and *m* is matrix (coffee or aqueous).

Thus, the suppression percentage is the mean of the suppression for each individual participant estimated by subtracting the evaluation of the pure solution from the evaluation of the mixed solution, where the taste stimuli are presented together, and then divided with the evaluation of the pure solution. Suppression Percentage calculations were performed using base R.

## 3. Results

### 3.1. Aqueous Matrix

Table 3 shows means and significant differences between samples for sweetness and sourness in the aqueous matrix sub-study. There was no significant difference between the control (CON) and the acid-containing samples (CA and TA) for sweetness. SU was highest for sweetness, followed by SUCA and SUTA. Thus sweetness was suppressed by acids.

Sucrose was found to suppress perceived sourness, as the highest perceived sourness was found for the acid-containing samples (CA and TA). The samples containing both sucrose and acid (SUCA and SUTA) were lower in perceived sourness and thus perceived sourness from the acids was suppressed by the sucrose. Perceived sourness was not significantly different between CON and SU.

**Table 3.** Means and HSD letters of differences in sweetness and sourness perception between aqueous samples. F- and *p*-values for ANOVA model. Letters are HSD pairwise significant, and are to be read row-wise. Samples are Control (CON), Citric Acid (CA), Tartaric Acid (TA), Sucrose (SU), Sucrose, and Citric Acid (SUCA), and Sucrose and Tartaric Acid (SUTA).

|  | CON | CA | TA | SU | SUCA | SUTA | F-Value | *p*-Value |
|---|---|---|---|---|---|---|---|---|
| Sweetness | 1.85 [a] | 2.10 [a] | 2.12 [a] | 6.97 [c] | 5.18 [b] | 4.98 [b] | 299.18 | <0.001 |
| Sourness | 1.80 [a] | 6.30 [c] | 6.19 [c] | 1.62 [a] | 4.49 [b] | 4.95 [b] | 241.64 | <0.001 |

Results from the ANOVA model for the aqueous solution can be seen in Table 4. The perceived sweetness was not only affected by sucrose, but also the acid in the matrix. Further, the two interacted, indicating that the acids affect the perceived sweetness of sucrose, not only by changing the sweetness of the matrix, but by suppressing the increase in perceived sweetness from adding sucrose. Similarly, perceived sourness was affected by both acid and sucrose. Also for perceived sourness did the two interact indicating that sucrose affects the perceived sourness of the acids by suppressing the increase in perceived sourness from the acids.

**Table 4.** *p*-values for ANOVA model of aqueous matrix.

|  | Sweetness | Sourness |
|---|---|---|
| Acid | <0.001 | <0.001 |
| Sucrose | <0.001 | <0.001 |
| Acid × Sucrose | <0.001 | <0.001 |

Generally, sucrose suppresses perceived sourness and acids suppress perceived sweetness. Savant and McDaniel [9] investigated perceived sourness suppression by the sweeteners sucrose, fructose, and glucose on sourness from citric acid, lactic acid, and malic acid. Their study indicated no significant differences between the sugars on perceived sourness suppression of citric acid at their low concentration condition with concentrations similar to those used in this study. Suppression effects of different acids were not directly compared but showed only minor differences that might not be significant. Our results are consistent with those findings. Even though we use other acids, we did not find differences between the effects on acids of a similar level of sourness in the aqueous matrix.

### 3.2. Coffee Matrix

In the coffee sub-study, evaluations of bitterness were included besides the sweetness and sourness attributes. All evaluations—sweetness, sourness and bitterness—for each coffee sample can be found in Table 5.

**Table 5.** Means and HSD letters of differences in sweetness, sourness and bitterness perception between coffee samples. F- and *p*-values for ANOVA model. Letters are HSD pairwise significant, and are to be read row-wise. Samples are Control (CON), Citric Acid (CA), Tartaric Acid (TA), Sucrose (SU), Sucrose, and Citric Acid (SUCA), and Sucrose and Tartaric Acid (SUTA).

|  | CON | CA | TA | SU | SUCA | SUTA | F-Value | *p*-Value |
|---|---|---|---|---|---|---|---|---|
| Sweetness | 3.18 [b] | 2.44 [a] | 2.56 [ab] | 6.20 [d] | 4.92 [c] | 5.32 [c] | 102.69 | <0.001 |
| Sourness | 3.64 [b] | 7.45 [d] | 7.23 [d] | 2.92 [a] | 6.99 [d] | 6.00 [c] | 126.94 | <0.001 |
| Bitterness | 5.68 [b] | 5.87 [b] | 5.94 [b] | 4.04 [a] | 4.67 [a] | 4.53 [a] | 19.27 | <0.001 |

SU was evaluated highest in perceived sweetness, followed by SUCA and SUTA. Of the non-sucrose-containing samples, CON and TA were highest in sweetness. This is similar to the results found for the aqueous solutions. In contrast to the aqueous matrix,

CA was evaluated lowest in sweetness of all coffee samples. This indicates suppression of a sweetness intrinsic to the coffee matrix, as no sucrose was added to CA and thus there was no sucrose to suppress.

Previous research has shown that the perceived sweetness of coffee is likely to originate from something else than sweet tasting compounds [36,37]. Batali et al. [36] suggest that the perceived sweetness of coffee is likely due to cross-modal interactions from aroma-taste interactions. Such interaction is presumably also the case in the present study.

Perceived sourness also differed in the coffee matrix compared to the aqueous matrix. Even though SUCA was lower than CA in perceived sourness, the difference does not reach significance. This can either be because there was no suppression of the perceived sourness of CA from the sucrose in coffee, or that it was lower than the suppression in aqueous solutions and therefore so low that it was not detected in this study. SUTA was evaluated lower than TA in perceived sourness, as was the case for the aqueous solutions. As SU was lower than CON, sucrose also suppressed the intrinsic sourness of the coffee. As will be presented later (Figure 1), the coffee CON was significantly more sour than the aqueous solution CON.

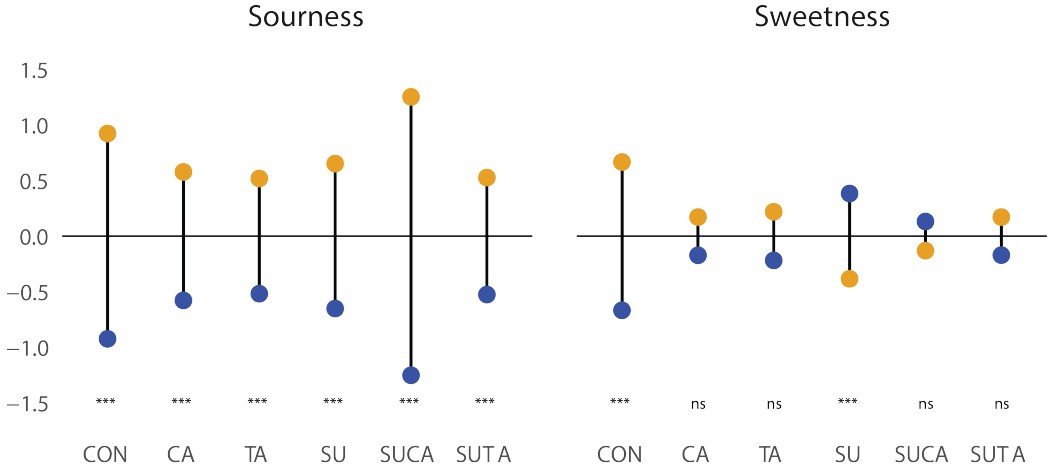

**Figure 1.** Deviations from mean for coffee (brown dots) and aqueous (blue dots) sub-studies, respectively. Significance: ns = not significant, *** < 0.001. Samples are Control (CON), Citric Acid (CA), Tartaric Acid (TA), Sucrose (SU), Sucrose, and Citric Acid (SUCA), and Sucrose and Tartaric Acid (SUTA).

Perceived bitterness was not affected by either citric or tartaric acid, but was significantly lower, and thus suppressed by sucrose, in all three sucrose-containing samples. Previous studies have found it difficult for naïve subjects to separate sour and bitter sensations, a so-called sour–bitter confusion [38–42]. This often leads to an increase in perceived bitterness when sourness is increased and vice versa. This was not seen in this study; at least the participants did not confuse the sourness from the acids with bitterness. It can, on the other hand, not be ruled out that the difference in perceived sourness between the CON in the aqueous matrix and the CON in the coffee matrix is at least partly due to a higher perceived bitterness in the coffee and not only due to a higher perceived sourness.

The varying and diminished suppression effects of both acid and sucrose described above are clear from the ANOVA results (Table 6). Neither for perceived sweetness nor perceived sourness is there an interaction effect from Acid × Sucrose, indicating no taste interactions between those in the coffee matrix. On the other hand, sucrose did affect perceived sourness independently of the acids as evident from the significant main effect of Sucrose on Sourness ($p < 0.001$). The acids also affected perceived sweetness independently of sucrose as evident from the main effect of Acid on Sweetness ($p < 0.001$).

**Table 6.** *p*-values for ANOVA model of coffee matrix.

|  | Sweetness | Sourness | Bitterness |
|---|---|---|---|
| Acid | <0.001 | <0.001 | 0.04 |
| Sucrose | <0.001 | <0.001 | <0.001 |
| Acid × Sucrose | 0.19 | 0.07 | 0.47 |

This means that there was a suppression effect of acids on perceived sweetness, though this effect was independent of the level of sucrose. The same was the case for the suppression of perceived sourness by sucrose which was also independent of the acid level. One way to interpret this is that taste will be suppressed as long as it is present, and this is independent of the added taste stimuli. This indicates that the intensity of the suppressed taste is less important when it comes to the degree of suppression, at least for the taste intensity differences investigated in this study. This would be true for both perceived sourness and perceived sweetness.

The results for the coffee matrix differ from the results for the aqueous matrix. In the coffee matrix, we saw no interaction effect between sucrose and acids for neither perceived sweetness nor perceived sourness. The sample differences for the perceived sourness of the acids with and without sucrose are similar to those found in the aqueous matrix, though citric acid suppresses the perceived sweetness of the coffee in contrast to tartaric acid.

The lower suppression effects of perceived sweetness and perceived sourness in the coffee matrix compared to the aqueous matrix are similar to the effects of matrix found by Zamora et al. [11] and could thus be expected.

*3.3. Comparisons*

To visually display any tendencies in perception of sweetness and sourness for the aqueous and coffee solutions, Figure 1 shows differences from the means of participants' evaluations of sweetness and sourness in the aqueous and coffee matrices for all samples. The most pronounced difference between the matrices is the difference in perceived sourness. For perceived sourness, all sample means differed between the matrices. Here, all coffee samples were considered more sour than the aqueous samples.

There is seemingly no systematic difference in perceived sweetness between the matrices. There is a significant difference between the control samples, showing higher perceived sweetness for coffee CON compared to aqueous CON. Further, perceived sweetness in SU is higher in the aqueous matrix than in the coffee matrix.

To quantify the taste interactions between perceived sweetness and perceived sourness, mean suppression percentages for each suppression and each matrix have been calculated (Table 7). This shows that much of the suppression diminished when presented in coffee compared to the aqueous matrix. This further confirms the findings from Sections 3.1 and 3.2 where suppression was higher in the aqueous matrix than in coffee.

**Table 7.** Mean suppression percentages for the different matrices. Only when the difference between the sample with and the sample without suppressant is significant is the suppression percentage shown. SwP is Percent Sweetness Suppression, SoP is Percent Sourness Suppression.

|  | Aqueous (%) | Coffee (%) |
|---|---|---|
| SwP by Citric Acid | 16 | 6 |
| SwP by Tartaric Acid | 14 | 3 |
| SoP from Citric Acid by Sucrose | 14 | - |
| SoP from Tartaric Acid by Sucrose | 9 | 6 |

The suppression percentages reveal small differences between acids. Generally, the percentages are higher for sweetness suppression by citric acid than by tartaric acid. This is also the case for sourness suppression by sucrose in an aqueous solution, where perceived

sourness of citric acid to a larger degree than that of tartaric acid is suppressed by sucrose. It seems to be the other way around in the coffee matrix, where the perceived sourness of tartaric acid is more suppressed than the perceived sourness of citric acid, which is not affected by sucrose.

Generally, the difference between the coffee matrix and the aqueous matrix seems to be two-fold. First, the suppression effect in coffee mostly seems to be related to factors intrinsic to the matrix whereas the effect in water is to a larger degree related to the added taste stimuli. Coffee contains many more taste and flavor compounds, and for sweetness, differences could at least partially be related to aroma-taste interactions [36,37]. For sourness it could be attributed either to acids in the coffee or to an interaction with bitterness. Second, the complex beverage matrix of coffee adds many different taste and flavor compounds which decrease the perceived differences between the samples.

## 4. Conclusions

Tastes interact with one another, and this study showed that sweetness was suppressed by both citric acid and tartaric acid in both an aqueous matrix and a coffee matrix. The suppression was lower in the coffee matrix than in the aqueous matrix. The sourness of both citric acid and tartaric acid was suppressed by sucrose in the aqueous matrix whereas only the sourness of tartaric, and not citric acid, was suppressed in the coffee matrix. In the coffee matrix, the sourness and sweetness of the matrix itself was to a large extend suppressed, whereas this was not seen in the aqueous matrix.

The suppression effects in the two beverages seem to relate to different factors. Where the suppression effect in coffee mainly seems to be related to interactions matrix-intrinsic taste and flavor compounds, the suppression effect in water seems to a larger degree related to the added taste stimuli. Further, the complex beverage matrix of cold brewed coffee adds many different taste and flavor compounds which seems to decrease the perceived differences between the samples.

These results show that, if done with caution, results from taste interaction studies conducted on aqueous matrices can to some extent be interpolated to more complex matrices, in this case coffee. However, to further substantiate these results, comparisons of other matrices are needed. Another important finding from this study is that, effects might diminish to some degree with the increase in matrix complexity.

**Author Contributions:** Conceptualization, J.Y.J., L.A.M. and U.K.; methodology, J.Y.J., L.A.M. and U.K.; software, J.Y.J.; validation, J.Y.J.; formal analysis, J.Y.J.; investigation, J.Y.J.; resources, U.K.; data curation, J.Y.J.; writing–original draft preparation, J.Y.J.; writing–review and editing, L.A.M., U.K.; visualization, J.Y.J.; supervision, L.A.M. and U.K.; project administration, J.Y.J. and U.K.; funding acquisition, U.K. All authors have read and agreed to the published version of the manuscript.

**Funding:** This work was supported by the Sino Danish Centre (SDC) within the 'Food and Health Research Theme', Aarhus, Denmark, the iFOOD Aarhus University Centre for Innovative Food Research, Aarhus, Denmark, and the Aarhus University Graduate School of Technological Sciences (GSTS), Aarhus, Denmark.

**Institutional Review Board Statement:** Ethical review and approval were waived for this study due to the reason that research in sensory properties of foods is exempted from the requirements of ethical approval. Participating in this study did not inflict risks beyond those encountered in normal everyday life.

**Informed Consent Statement:** Informed consent was obtained from all subjects involved in the study.

**Data Availability Statement:** The data presented in this study are available on request from author.

**Acknowledgments:** The authors would like to thank Line Elgaard Nielsen, Jens Madsen, Ying Bai, and Lucie Bauderlique for assistance in conducting the study aqueous study, and Julie Moustgaard, Linea Sandgren and Signe Lund Mathiesen for conducting the coffee sub-study.

**Conflicts of Interest:** The authors declare no conflict of interest.

## Abbreviations

The following abbreviations are used in this manuscript:

| | |
|---|---|
| CON | Control sample |
| SU | Sucrose sample |
| CA | Citric acid sample |
| TA | Tartaric acid sample |
| SUCA | Sucrose and citric acid sample |
| SUTA | Sucrose and tartaric acid sample |
| HSD | Tukey's Honest Significant Differences test |
| SwP | Suppression Percentage for sweetness |
| SoP | Suppression Percentage for sourness |
| MDPI | Multidisciplinary Digital Publishing Institute |
| DOAJ | Directory of open access journals |

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
