# Peer review of "Comparison of Sweet–Sour Taste Interactions between Cold Brewed Coffee and Water"

_beverages, doi:10.3390/beverages8020032_

Round 1
Reviewer 1 Report
1. This article investigated the taste suppression between sweetness and sourness. The finding is complex matrix decrease the interaction. There were large number of people joined the sensory evaluation, the data is solid.
2. Because of the complex of coffee matrix it is hard to evaluate the taste interaction in matrix. Can the author add some description about why use coffee as a matrix, what is the advantage of coffee matrix? in the introduction part.
3. English may be need checked. e.g. line 43, most often taste interaction studies; line 44, Besides……; line 45, Besides……;
4. 3.1-3.2 and table 3,4,5,6 does not add any new things to know. Basically it was flavor perception interaction in water. If there is some thing new to say, please point out in section 3.1-3.2.
5. In conclusion, suppression effect in coffee seems to mostly related to matrix intrinsic factors. What is these factors? Is there any discussion or mechanism to explain the intrinsic factors for suppression?
Reviewer 2 Report
Dear Editor,
The paper presented by Jonas Y de Junge et collaborators reveals an interesting approach of matrix beverages.
The work is well presented and structured. The experiments have been very well designed and carried out and the conclusions achieved on the basis of the experimental data are good justified and very useful. The methods used in this study are specific and the experimental is very well explained.
The paper could be read by a native speaker to ensure the correct use of English in a few points, however, there are no major grammatical errors that would make the text difficult to comprehend.
I suggest that authors should mentioned the status of the health participants (on their own statement), because if they have injuries/ diseases that could affect the response, the study is not relevant.
Round 2
Reviewer 1 Report
The revision shows significant improve. The content is concise after rewritten. There is no more question for this article.